# Infection with *Helicobacter pylori* Induces Epithelial to Mesenchymal Transition in Human Cholangiocytes

**DOI:** 10.3390/pathogens9110971

**Published:** 2020-11-21

**Authors:** Prissadee Thanaphongdecha, Shannon E. Karinshak, Wannaporn Ittiprasert, Victoria H. Mann, Yaovalux Chamgramol, Chawalit Pairojkul, James G. Fox, Sutas Suttiprapa, Banchob Sripa, Paul J. Brindley

**Affiliations:** 1Research Center for Neglected Tropical Diseases of Poverty, Department of Microbiology, Immunology and Tropical Medicine, School of Medicine & Health Sciences, The George Washington University, Washington, DC 20037, USA; aquiver_@hotmail.com (P.T.); skspeight08@gmail.com (S.E.K.); wannaporni@gwu.edu (W.I.); vmann@gwu.edu (V.H.M.); 2Tropical Disease Research Laboratory, Faculty of Medicine, Khon Kaen University, Khon Kaen 40002, Thailand; sutasu@kku.ac.th; 3Department of Pathology, Faculty of Medicine, Khon Kaen University, Khon Kaen 40002, Thailand; cyaova@yahoo.com (Y.C.); chawalit-pjk2011@hotmail.com (C.P.); 4Division of Comparative Medicine, Massachusetts Institute of Technology, Cambridge, MA 02139, USA; jgfox@mit.edu

**Keywords:** *Helicobacter pylori*, cholangiocyte, epithelial-to-mesenchymal transition

## Abstract

Recent reports suggest that the East Asian liver fluke infection, caused by *Opisthorchis viverrini*, which is implicated in opisthorchiasis-associated cholangiocarcinoma, serves as a reservoir of *Helicobacter pylori*. The opisthorchiasis-affected cholangiocytes that line the intrahepatic biliary tract are considered to be the cell of origin of this malignancy. Here, we investigated interactions in vitro among human cholangiocytes, *Helicobacter pylori* strain NCTC 11637, and the congeneric bacillus, *Helicobacter bilis*. Exposure to increasing numbers of *H. pylori* at 0, 1, 10, 100 bacilli per cholangiocyte of the H69 cell line induced phenotypic changes including the profusion of thread-like filopodia and a loss of cell-cell contact, in a dose-dependent fashion. In parallel, following exposure to *H. pylori*, changes were evident in levels of mRNA expression of epithelial to mesenchymal transition (EMT)-encoding factors including snail, slug, vimentin, matrix metalloprotease, zinc finger E-box-binding homeobox, and the cancer stem cell marker CD44. Analysis to quantify cellular proliferation, migration, and invasion in real-time by both H69 cholangiocytes and CC-LP-1 line of cholangiocarcinoma cells using the xCELLigence approach and Matrigel matrix revealed that exposure to ≥10 *H. pylori* bacilli per cell stimulated migration and invasion by the cholangiocytes. In addition, 10 bacilli of *H. pylori* stimulated contact-independent colony establishment in soft agar. These findings support the hypothesis that infection by *H.*
*pylori* contributes to the malignant transformation of the biliary epithelium.

## 1. Introduction

There is increasing evidence that the East Asian liver fluke *Opisthorchis viverrini* may serve as a reservoir of *Helicobacter*, which, in turn, implicates *Helicobacter* in the pathogenesis of opisthorchiasis-associated cholangiocarcinoma (CCA) [1,2,3,4,5]. The International Agency for Research on Cancer of the World Health Organization categorizes infection with the food-borne liver flukes, *O. viverrini* and *Clonorchis sinensis*, as Group 1 carcinogens [6]. In a similar fashion, infection with *Helicobacter pylori* is categorized as a Group 1 carcinogen [6]. In northern and northeastern Thailand and Laos, opisthorchiasis is the major documented risk factor for CCA [6,7,8,9]. Given the elevated prevalence of CCA in this region, where infection with liver fluke prevails, and given the evidence of linkages between infection by *Helicobacter* during opisthorchiasis, these two biological carcinogens together may orchestrate the pathogenesis of opisthorchiasis and bile duct cancer. Indeed, it has been hypothesized that the association of *Helicobacter* and its virulence factors, including during opisthorchiasis, underlies much biliary tract disease including CCA in liver fluke-endemic regions [10]. Colonization by species of *Helicobacter* causes hepatobiliary disease that can resemble opisthorchiasis [5,11,12,13,14]. Lesions ascribed to liver fluke infection including cholangitis, biliary hyperplasia and metaplasia, periductal fibrosis, and CCA may be, in part, *Helicobacter*-associated hepatobiliary disease.

With respect to the gastric epithelium and the association with stomach adenocarcinoma, *H. pylori* colonizes the mucosal layer [6] and adheres to the epithelium through bacterial adhesins with cellular receptors [15], from where its virulence factors stimulate a cascade of inflammatory signaling, anti-apoptosis, cell proliferation, and transformation pathways by its virulence factors [16,17,18]. Fox and coworkers were the first to speculate that *H. pylori* also can cause hepatobiliary diseases in humans [19]. This is the dominant species among the genus *Helicobacter* detected in the context of hepatobiliary disease [20] and *H. pylori* is detected more frequently in cases with CCA or hepatocellular carcinoma (HCC) than in those with benign tumors and other control groups [21,22], suggesting a positive correlation between *H. pylori* infection and hepatic carcinogenesis. In addition to the extensive literature on the interactions between *H. pylori* and gastric cells [17,23], interactions between biliary epithelium and this bacillus have been reported. Among these, in vitro studies revealed that *H. pylori* induces multiple effects in CCA cell lines, including inflammation (IL-8 production), cell proliferation, and apoptosis. Even at low multiplicity of infection, *H. pylori* induces pro-inflammatory cytokine and cell proliferative responses in CCA cell lines [4,24], and even small numbers of bacilli that likely reach the biliary tract routinely may be sufficient to promote inflammation and transformation of the biliary epithelia [24].

Commensalism involving *H. pylori* and *O. viverrini* may have evolved and may facilitate conveyance of the bacillus into the biliary tract during the migration of the juvenile fluke following ingestion of the metacercaria in raw or undercooked cyprinid fish and establishment of liver fluke infection [1,25]. Since curved rods resembling *Helicobacter* have been documented in the digestive tract of *O. viverrini* [1] and, given the low pH of the gut of the fluke, the *H. pylori* rods or spores might be transported from the stomach to the duodenum by the migrating larval parasite. The curved, helical *H. pylori* rod attaches to a cholangiocyte, which internalize in a similar fashion to its colonization of mucous-producing cells of the gastric epithelia [17,26,27].

Transition of epithelial cells to mesenchymal cells during disease and epithelial to mesenchymal transition (EMT) during development and wound healing follow evolutionary conserved routes with well-characterized morphological and other phenotypic hallmarks [28]. These phenotypes are showcased during the malignant transformation of the gastric mucosa, resulting from infection with *H. pylori* [29]. Here, we investigated interactions between the NCTC 11637 strain of *H. pylori*, the related species *H. bilis* [30], and the human cholangiocyte. Colonization by *H. pylori* induced EMT, cellular proliferation, cell migration, and invasion responses, and induced wound closure by cells of the H69 human cholangiocyte line and also by cells of the CC-LP-1 line of cholangiocarcinoma cells [31,32,33].

## 2. Results

### 2.1. Helicobacter Pylori Induces Epithelial to Mesenchymal Transition in Cholangiocytes

H69 were directly exposed to increasing number of *H. pylori* strain NCTC 11637 at 0, 10, and 100 bacilli per cholangiocyte in 6-well plates. Twenty-four hours after the addition of the bacilli, the epithelial cell appearance had altered to an elongate phenotype characterized by terminal thread-like filopodia and diminished cell-to-cell contacts (Figure 1A–C). The extent of the transformation was dose-dependent and, also, it indicated increased motility by the cholangiocytes. The morphological change was quantified by assessment and measurement of the length to width ratio of the H69 cells. The ratio increased significantly at both an MOI of 10 and of 100 when compared with an MOI of 0 (*p* ≤ 0.01) (Figure 1D–F). In addition, the numbers of isolated, single H69 cells increased significantly to the increasing length to width ratio of the *H. pylori*-exposed cholangiocytes, *p* ≤ 0.05 and ≤0.001 at an MOI of 10 and 100 *H. pylori*, respectively (Figure 1G). The cholangiocytes displayed a hummingbird phenotype-like appearance, reminiscent of gastric adenocarcinoma cells [29]. Moreover, compared to non-treated control cells, H69 cells cultured in medium supplemented with TGF-β at 5 ng/mL displayed cellular aggregation and nodule formation within 24 h (Appendix A).

### 2.2. EMT-Associated-Factors Induced by Exposure to H. pylori

Transcriptional dynamics of the apparent EMT was investigated using qRT-PCR of total cellular RNAs for six EMT-related factors including the mesenchymal marker vimentin, the transcription factors, Snail, Slug, and ZEB1, the adhesion molecule JAM1, and the proteolytic enzyme, MMP7 [34,35,36]. The cancer stem cell markers, CD24 and CD44 [37], were also monitored. Levels of the fold-difference in the transcription for each of the six markers increased in a dose-dependent fashion. The regulatory transcriptional factor, Snail was markedly up-regulated with the highest level of fold difference, namely, 88-, 656-, and 5309-fold at an MOI of 10, 50, and 100, respectively, followed by MMP7, with 15-, 21-, and 44-fold at an MOI of 10, 50, and 100, respectively. Transcription of Slug, ZEB1, vimentin, and JAM1 also was up-regulated during bacterial colonization. CD44 was up-regulated, whereas CD24 expression was not changed significantly, revealing a CD44^+^ high/CD24^+^ low phenotype in cells exposed to *H. pylori* (Figure 2; *p* ≤ 0.05 to ≤0.0001 for individual EMT markers and/or MOI level, as shown). A pattern of CD44^+^ high/CD24^+^ low expression is a cardinal character of cancer stem cell activity in gastric adenocarcinoma [37].

### 2.3. Exposure of Cholangiocytes to H. pylori Induces Cellular Migration, Invasion, and Wound Closure

Analysis in real time using a Boyden chamber-type apparatus revealed that exposure to *H. pylori* at an MOI of 10 to 50 *H. pylori* significantly stimulated migration of H69 cells from 24 to 96 h after starting the assay (Figure 3A; *p* values at representative times and MOIs). However, the effect was not evident at an MOI of 100.

A similar assay was undertaken using the CC-LP-1 cholangiocarcinoma cell line but at a lower MOI. At an MOI of 1, 10, and 50, *H. pylori* stimulated significantly more migration of CC-LP-1 cells from 20 to 40 h after starting the assay (Figure 3B). Concerning the invasion of the extracellular matrix, both the CC-LP-1 and H69 cells migrated and invaded the Matrigel layer in the UC of the CIM plate at significantly higher rates than did the control cells, with significant differences evident from 48 to 96 h (Figure 3C,D). In addition, scratch assays revealed two-dimensional migration of H69 cells over 24 h. Wound closure by H69 cells increased significantly to 19.47% at MOI of 10 (*p* ≤ 0.05) although an effect at an MOI of 100 was not apparent in comparison with the control group (Figure 4).

### 2.4. H. pylori Induces Anchorage-Independent Colony Formation

As a prospective biomarker of malignant transformation of the *Helicobacter*-infected cholangiocytes, anchorage-independent colony formation by the cells was determined in a medium of soft agar. Following maintenance of the cultures for up to 28 days, counts of the number of colonies was revealed that an MOI of 10 *H. pylori* had induced a significant increase in colony numbers of the H69 cells (Figure 5A,B). Significant differences from the control group were not seen at an MOI of 50 or 100. The size of colonies also increased in all groups exposed to *H. pylori*, although this was statistically significant only at an MOI of 50 (Appendix A; *p* ≤ 0.05). By contrast to *H. pylori*, exposure of H69 cells to *H. bilis*, a microbe that naturally resides in the mammalian biliary tract, intestines, and other sites [38,39], decreased the number of colonies that formed in soft agar (Figure 5C). The neutral or even inhibitory effect of *H. bilis* on H69 was mirrored by the lack of cellular proliferation by H69 cells when monitored in real-time over 72 h at an MOI of 0, 10, 50, and 100 *H. bilis* (Appendix A).

## 3. Discussion

Numerous species of *Helicobacter* have been described [40], while *H. pylori* was the first prokaryote confirmed to cause gastric disease, including peptic ulcer, gastric mucosa-associated tissue lymphoma, and adenocarcinoma [41,42,43,44,45]. Bacilli of *H. pylori* occur in the stomach in at least half of the human population. Transmission is from mother to child and by other routes. The human-*H. pylori* association may be beneficial in early life, including contributions to a healthy microbiome and reduced risk of early-onset asthma [46,47]. Nonetheless, diverse disease outcomes have been ascribed to *H. pylori*, and these are known to vary in distinct geographical regions. An increasing complement of virulence genes and genotypes have been reported including virulence genes specifically involved with adhesion and colonization of the gastric epithelium, injury to the mucosae, and proinflammation and immune responses [48,49]. Major virulence genes include the cytotoxin associating gene protein (CagA), the vacuolating cytotoxin A (VacA) [50], the outer inflammatory protein (OipaA) [51], the TNF-α-inducing protein (Tip-α) [52], and many others [53].

Here, we investigated interactions among a human cholangiocyte cell line, H69, a CCA cell line CC-LP-1, CagA^+^
*H. pylori*, and *H. bilis* bacilli. Infection of H69 with *H. pylori* induced EMT in a dose-dependent manner, which was characterized by cell elongation and scattering, which, in turn, implicate increasing change in cell motility. This visible appearance of these infected H69 cells resembled the hummingbird phenotype of gastric epithelial cells after exposure to *H. pylori* [28,29,54]. In the AGS cell line, a human gastric adenocarcinoma cell line [55], delivery of CagA by the type IV secretion mechanism from *H. pylori* subverts the normal signaling leading to actin-dependent morphological rearrangement. The hitherto uniform polygonal shape becomes markedly elongated with terminal needle-like projections [56].

AGS cells infected with *H. pylori* demonstrating the hummingbird phenotype also display early transcriptional changes that reflect EMT [28]. Here, H69 cells exposed to *H. pylori* exhibited an up-regulation of the expression of Snail, Slug, vimentin, JAM1, and MMP7 in a dose-dependent fashion, changes that strongly supported the EMT of this informative cholangiocyte cell line. Likewise, Snail, an *E*-cadherin repressor, was markedly up-regulated, which indicated that this factor may be a key driver of EMT in cholangiocytes. CD44 expression was up-regulated in dose-dependent fashion, whereas CD24 was not significantly changed, revealing a CD44^+^/CD24^−/low^ phenotype following infection. Colonization by *H. pylori* may not only induce EMT but may also contribute to stemness and malignant transformation of the cholangiocyte [37,57].

Realtime cell monitoring revealed that the H69 and CC-LP-1 cells proliferated, and migrated in response to *H. pylori*. Moreover, 10 bacilli of *H. pylori* per biliary cell stimulated cellular migration and invasion by both H69 and CC-LP-1 through an extracellular basement matrix, a behavioral phenotype also characteristic of EMT. The responses by CC-LP-1 were even more marked than those of H69, which is not surprising given that CC-LP-1 is a cancer cell line. In addition, following exposure to *H. pylori*, H69 cells accomplished anchorage-independent cell growth in soft agar, a phenotype indicative of the escape from anoikis and characteristic of metastasis [58]. The numbers of colonies of H69 cells in soft agar increased significantly following exposure to *H. pylori* at MOI of 10. By contrast, observations of contact independent cell growth and also growth responses by H69 cells to *H. bilis* as monitored and quantified using the RTCA approach, revealed that both the numbers of colonies of H69 in soft agar and cell growth during the RTCA assay decreased. The responses indicated that *H. pylori* strain NCTC 11637, but not *H. bilis*, displays the potential to induce neoplastic changes in cholangiocytes in a similar fashion to the gastric epithelia. Overall, the pro-carcinogenic changes induced by *H. pylori* but not by *H. bilis* were not unexpected, given that *H. pylori* is a biological carcinogen, even though *H. bilis* colonized the intestines and hepatobiliary tract, and has been associated with hepatobiliary disease including multifocal chronic hepatitis [39] and biliary tract malignancies [59]. Lastly, despite a clear stimulatory impact on the cholangiocytes of exposure of *H. pylori*, the physiological response was generally dose-dependent; cellular migration, wound healing, and contact-independent colony growth were impeded or even blocked at MOI of 50 or greater compared to enhanced growth stimulated by an MOI of 10. Dose-dependent responses as well as negative impacts at a higher MOI and/or higher concentrations of virulence factors of *H. pylori* have been reported during infection with *H. pylori* [60,61,62,63,64].

In addition to the well-known association with gastric cancer, *H. pylori* has been associated with hepatobiliary disease [4,5,65]. Related species of *Helicobacter*, *H. hepaticus*, and *H. bilis* are also associated with hepatobiliary disease [65,66,67]. Infection with the fish-borne liver flukes, *Opisthorchis viverrini* and *Clonorchis sinensis*, is classified as a Group 1 carcinogen by the International Agency for Research on Cancer, as is infection with *H. pylori* [6]. Opisthorchiasis is a major risk factor for cholangiocarcinoma in northeastern Thailand and Laos [6,7,8,9]. In addition to gastric disease, infection with species of *Helicobacter* causes hepatobiliary tract diseases that can resemble opisthorchiasis [5,8,9,11,12]. Liver fluke infection can induce lesions in the biliary system, including cholangitis, biliary hyperplasia and metaplasia, periductal fibrosis, and CCA. These lesions derive not only from liver fluke infection but are perhaps in part the consequence of hepatobiliary tract infection with *H. pylori*. *Helicobacter* may transit from the stomach to the duodenum and enter the biliary tree through the duodenal papilla and ampulla of Vater [26,68], and indeed, may be vectored there by the liver fluke, *O. viverrini* [1,2,3]. *H. pylori*-specific DNA sequences have been detected in CCA tumors and also from lesions diagnosed as cholecystitis/cholelithiasis in regions endemic for opisthorchiasis [5,11]. Furthermore, serological findings have implicated infection with *H. pylori* as a risk for CCA in Thailand [12].

The present findings supported the hypothesis that opisthorchiasis and concomitant colonization by *H. pylori* together may hasten or even synergize the malignant transformation of cholangiocytes [2,3,69]. By contrast, co-infections of *Helicobacter* species, including *H. pylori* and some other helminths, have generally been associated with a diminished risk of *H. pylori*-associated gastric carcinoma [70,71,72]. Notably, concurrent infection in mice with an intestinal nematode that modulates inflammation induces a Th2-polarizing cytokine phenotype with concomitant downmodulation of Th1 and the gastric immune responses, and reduces *Helicobacter*-induced gastric atrophy [73]. Nonetheless, given that exposure to *H. pylori* NCTC 11637 stimulated EMT in cholangiocytes and which, in turn, also suggests a role in the underlying fibrosis [74,75,76] and metastasis [77,78,79] of cholangiocarcinoma, we now might have a clearer explanation for why infection with the liver fluke induces CCA—involvement by *H. pylori* and its virulence factors [80,81].

Our report has some limitations. This study of the colonization of the human cholangiocyte by *H. pylori* was undertaken with the NCTC 11637 strain of *Helicobacter pylori*, a CagA positive infraspecific strain of the pathogen. We observed that exposure of H69 cells to *H. pylori* induced EMT, cell proliferation and invasion, wound closure, and other responses. The report provides novel, informative findings, but it is preliminary in scope with respect to the investigation of the impact of *H. pylori* on the biliary tract. Deeper investigation using other genotypes of *H. pylori* along with cognate deletion mutants would be appropriate in future studies in order to define specific virulence factors of *H. pylori* that underpin the phenotypic responses described here. The *H. pylori* strain NCTC 11637 is CagA positive, and given the established and well-characterized role of CagA in malignant transformation of the gastric epithelium [29,82,83,84], assays that include additional CagA-positive strains such as P12 [85], and the inclusion of a CagA negative and/or *cag*PAI-dysfunctional strain such as SS1 [86,87,88], would explore the specific contribution of the CagA virulence factor to the EMT and other responses observed in the H69 line cholangiocyte. Specifically, the induction of the hummingbird phenotype in gastric epithelial cells by CagA is dependent on the number and type of its EPIYA phosphorylation site repeats [83,89]. Moreover, to confirm that the CagA virulence factor of *H. pylori* strain NCTC 11637 induced the elongate, hummingbird-like appearance of the H69 cholangiocyte, a demonstration that H69 cells could also be infected by another CagA positive *H. pylori*, and that the *cag*PAI cag pathogenicity island encoded type IV secretory system was functional and active during the infection, as established by detection of phosphorylated CagA [90,91,92], would be required.

Although *H. pylori* strain NCTC 11637 is CagA positive, other virulence factors of *H. pylori* may underlie the phenotypic changes in the H69 cells given that some other virulence factors *of H. pylori* induce phenotypic changes in gastric epithelia similar to those observed here in cholangiocytes. Notably, the outer membrane antigen, lipoprotein 20 (Lpp20) [93], promotes cell migration and down-regulation of *E*-cadherin in gastric cancer cells, responses involved with EMT [94]. Lpp20 also stimulates cell proliferation. As with the investigation of the role of CagA discussed above, a similar investigation might be undertaken to confirm a role for Lpp20, and indeed for other virulence factors.

To conclude, the central finding of this report was that colonization by *H. pylori* induced epithelial to mesenchymal transition in human cholangiocytes. Further investigation of the relationship between *O. viverrini* and *H. pylori* within the infected biliary tract is warranted. Studies on the tumorigenicity of the *H. pylori*-transformed H69 cells in immune-suppressed mice would likely be informative.

## 4. Materials and Methods

### 4.1. Cell Lines of Human Cholangiocytes

The immortalized intrahepatic cholangiocyte cell line, H69 [31,32], Cellosaurus [95] identifier RRID:CVCL_812,1, and the primary cholangiocarcinoma cell line, CC-LP-1 [33,96] Cellosaurus RRID CVCL_0205, were obtained as described [33,96,97,98]. In brief, H69 cells were cultured in Dulbecco’s Modified Eagle’s Medium (DMEM) (Thermo Fisher Scientific, Inc., Waltham, MA, USA), DMEM/F12 (Millipore-Sigma, St. Louis, MO, USA) supplemented with 10% fetal bovine serum (FBS) (Invitrogen, Carlsbad, CA, USA; Thermo Fisher Scientific, Inc.) and adenine, insulin, epinephrine, triiodothyronine/transferrin, hydrocortisone, epidermal growth factor, and penicillin/streptomycin (all from Millipore-Sigma, Burlington, MA, USA), as described [97]. CC-LP-1 cells were cultured in DMEM containing 10% FBS, L-glutamine, and penicillin/streptomycin. Both cell lines were maintained at 37 °C in humidified 5% CO_2_ in air. The cells were cultured to ~80% confluence before co-culture with *H. pylori*. H69 cells at passages 10 to 22 only and CC-LP-1 cells at passages five to 10 were used in this study.

### 4.2. Helicobacter pylori and Helicobacter bilis

*Helicobacter pylori* NCTC 11637 = CCUG 17874 = ATCC 43504 (Epsilon-proteobacteria), infraspecific name, strain NCTC 11637, from human gastric antrum [45], and *Helicobacter bilis* (ATCC^®^ 49314™), originally from aborted sheep fetus, liver, and fluids, from Brookings, South Dakota [38,99,100], were purchased from the American Type Culture Collection (ATCC) (Manassas, VA). Both were maintained on trypticase soy agar with 5% sheep blood for 72 to 96 h [101] (Becton, Dickinson and Company, Franklin Lakes, NJ, USA) under a microaerobic atmosphere established with the BD GasPak^™^ EZ container system and reagents (Becton Dickinson). The GasPak container with the plates inoculated with the bacteria was located within an orbital shake (MaxQ™ 4000 Benchtop Orbital Shaker, Thermo Fisher) and incubated at 37 °C for 96 h. At this point, each species of *Helicobacter* was harvested by scraping the colonies from the agar plate, and scrapings of the culture on the agar pale were resuspended in DMEM/F12 medium until an optical density at 600 nm of 1. 0 was reached, which corresponds to ~1 × 10^8^ colony forming units (cfu)/mL [102,103,104]. Viability of the bacteria was confirmed by visual inspection for movement of the flagellated bacteria.

### 4.3. Morphological Assessment of Cholangiocytes

H69 cells were co-cultured with *H. pylori* at increasing multiplicities of infection (MOIs) of 0 (no bacilli), 10, and 100 for 24 h in serum-free media. After 24 h, images of the cells were captured using a digital camera fitted to an inverted microscope (Zeiss Axio Observer A1, Jena, Germany) in order to document the morphology of the *H. pylori*-exposed cholangiocytes.

### 4.4. Cell Scattering and Elongation

H69 cells were seeded at 5 × 10^5^ cells/well in 6-well culture plates (Greiner Bio-One, Monroe, NC, USA). At 24 h, the culture medium was exchanged for serum- and hormone-free medium containing *H. pylori* at an MOI of 0, 10, and 100, respectively, and maintained for a further 24 h. At that interval, the appearance, including scattering of the cells in the culture, was documented as above. Cell scattering was quantified by counting the total number of isolated, single cells per field in 10 randomly selected images at 5× magnification [105]. To assess elongation of cells, images were documented of the cells in ~20 randomly selected fields of view at ×20 magnification, with two to seven cells per field. The length to width ratio of isolated cells was ascertained using measurement tools in the ImageJ software [106].

### 4.5. In Vitro Wound Healing Assay

Monitoring cell migration in a two-dimensional confluent monolayer may facilitate characterization the process of wound healing with respect with exposure to *H. pylori* [107]. A sheet migration approach was used to assay wound closure [97,108,109] following exposure of the cholangiocytes to *H. pylori*. H69 cells infected with *H. pylori* at an MOI of 0, 10, and 100 were cultured overnight in 6-well plates to allow cell adherence. A linear scratch to wound the monolayer was inflicted with a sterile 20 µL pipette tip. The dimensions of wound were documented at 0 and at 26 h, and the rate of wound closure quantified by measurement of the width of the wound in the experimental and the control (MOI of 0) groups of cells [97,98].

### 4.6. Real Time Quantitative PCR (RT-qPCR)

Total RNAs from H69 cells were extracted with RNAzol (Molecular Research Center, Inc., Cincinnati, OH, USA) according to the manufacturer’s instructions. The RNA Quantity and quality of the RNAs were established by spectrophotometry (NanoDrop 1000, Thermo Fisher Scientific). Total RNAs (500 ng) were reversed-transcribed using the iScript cDNA Synthesis Kit (Bio-Rad, Hercules, CA). Analysis of expression of six EMT-associated genes—vimentin, snail, slug, ZEB1, JAM1, and MMP7, and two cancer stem cell markers, CD44 and CD24—was undertaken by quantitative reverse transcription-polymerase chain reaction (qRT-PCR) of total RNAs using an ABI7300 thermal cycling system (ABI) and the SsoAdvance SYBR green mixture (Bio-Rad). Signals were normalized to expression levels for GAPDH. The relative fold-change was determined by the ΔΔCt method [110]. Three biological replicates of the assay were undertaken.

Appendix A provides the oligonucleotide sequences of the primers in the RT-qPCRs. The design of these primers for the human EMT and stem cell markers was undertaken used Primer-BLAST, https://www.ncbi.nlm.nih.gov/tools/primer-blast/index.cgi [111], with the genome of *Helicobacter* specifically excluded during the blast search. Furthermore, before the RT-qPCR analysis was undertaken, conventional PCR was carried out using cDNA from H69 cells as the template along with these primers. Products were sized using ethidium bromide-stained agarose gel electrophoresis, which confirmed the presence of amplicons of predicted sizes (not shown).

### 4.7. Assessment of Cell Proliferation

Cell proliferation of H69 was assessed using the xCELLigence real-time cell analyzer (RTCA) DP system (ACEA Biosciences, San Diego, CA, USA), as described [97,112,113,114]. H69 cells were fasted for 4 to 6 h in 1:20 serum diluted medium, 0.5% FBS final concentration, as described [115,116], after which cells were co-cultured of viable bacilli of *H. pylori* for 120 min before starting the assay. H69 cells were subsequently harvested in 0.25% trypsin-EDTA and then washed with medium. Five thousand H69 cells were seeded on to each well of the E-plate in H69 medium and cultured for 24 h. The culture medium was removed, and the cells were gently washed with 1 × PBS. The PBS was replaced with serum diluted medium (above). The cellular growth was monitored in real time with readings collected at intervals of 20 min for ≥60 h. For quantification, the cell index (CI) [113] was averaged from three independent measurements at each time point.

### 4.8. Cell Migration and Invasion in Response to H. pylori

To monitor the rate of cell migration and/or invasion in real time, we used the xCELLigence DP instrument (above) equipped with a CIM-plate 16 (Agilent ACEA Biosciences, San Diego, CA, USA), which is an electronic Boyden chamber consisting of a 16-well culture plate in which each well includes an upper chamber (UC) and a lower chamber (LC) separated by a microporous (pore diameter, 8 µm) membrane. In response to a chemoattractant, cells may migrate/invade from the UC towards the membrane and the LC. Migrating cells come into contact with and adhere to microelectronic sensors on the underside of the membrane that separates the UC from the LC, leading to a change in the relative electrical impedance [113,114,117]. To investigate the migration, H69 and CC-LP-1 cells were infected with *H. pylori* at MOIs of 0, 1, 10, 50, and 100, respectively, for 120 min before the start of the assay, after which the cells were harvested following treatment for 3 min with 0.25% trypsin-EDTA and washed with medium. Subsequently, 30,000 H69 or CC-LP-1 cells were seeded into the UC of the CIM-plate in serum-free medium. Complete medium (including 10% FBS) was added to the LC. Equilibration of the loaded CIM-plate was accomplished by incubation at 37 °C for 60 min to allow cell attachment, after which monitoring for migration commenced in real time for a duration of ~96 h with readings recorded at 20 min intervals. For quantification, CI was averaged from three independent measurements at each time point [118].

To investigate invasion, H69 or CC-LP-1 cells were infected with *H. pylori* at either MOI of 0 or 10 at 120 min before commencing the assay. The base of the UC of the CIM plate was coated with 20% solution of a basement membrane matrix (BD Matrigel, BD Biosciences, San Jose, CA, USA) in serum-free medium. A total of 30,000 of H69 or CC-LP-1 cells were dispensed into the UC in serum-free medium. The LC was filled with H69 or CC-LP-1 medium, as appropriate, containing 10% FBS. The loaded CIM-plate was sealed, inserted into the RTCA DP xCELLigence instrument at 37 °C in 5% CO_2_ and held for 60 min to facilitate attachment of the cells, after which electronic recording was commenced. Monitoring of invasion in real time continued for ~96 h with the impedance value recorded at intervals of 20 min continuously during the assay. CI was quantified as above.

### 4.9. Colony Formation in Soft Agar

For the soft agar assay, H69 cells were infected with *H. pylori* or *H. bilis* at an MOI of 0, 10, 50, and 100 at the start of the assay. H69 cells at 10,000 cells per well were mixed with 0.3% NuSieve GTG low melting temperature agarose (Lonza Pharma & Biotech, Walkersville, MD, USA) in complete medium, and plated on top of a solidified layer of 0.6% agarose in growth medium, in wells of 6- or 12-well culture plates (Greiner Bio-One, Monroe, NC, USA). Fresh medium was added every three or four days for 28 days or until the colonies had established, at which time the number of colonies of ≥50 µm diameter was documented using a camera fitted to an inverted microscope (above), and quantified [119].

### 4.10. Statistical Analysis

Differences in cell elongation, cell scattering, wound healing, and colony formation among experimental and control groups were analyzed by a one-way analysis of variance (ANOVA). Fold differences in mRNA levels, and real time cell migration and invasion through extracellular matrix were analyzed by a two-way ANOVA followed by Dunnett’s test for multiple comparisons. Three or more replicates of each assay were carried out. The analysis was conducted using GraphPad Prism v7 (GraphPad, San Diego, CA, USA). A *p* value of ≤0.05 was considered to be statistically significant.

## Figures and Tables

**Figure 1 pathogens-09-00971-f001:**
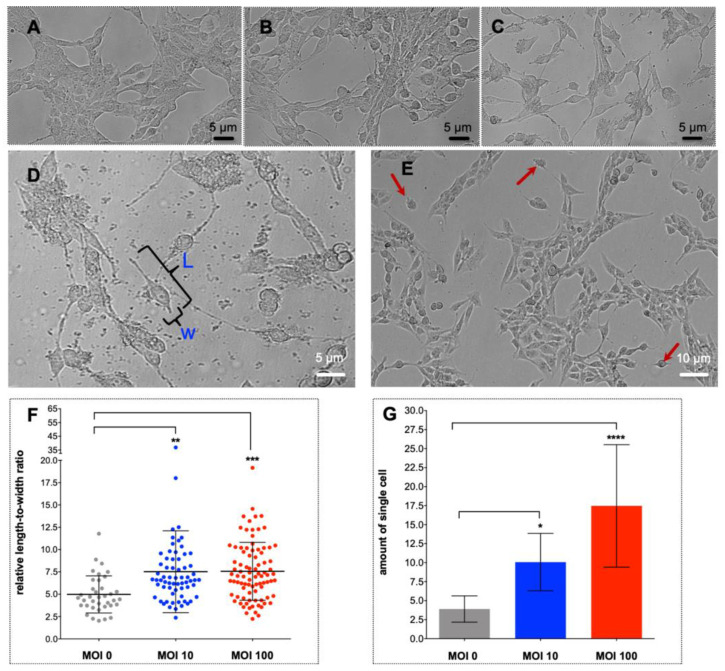
Exposure to CagA^+^
*Helicobacter pylori* strain NCTC 11637 (ATCC 43504) induced morphological alteration in cholangiocytes that included cellular elongation, terminal thread-like filopodia and diminished cell-to-cell contacts, indicative of epidermal to mesenchymal transition. Panels **A**–**C**: photomicrographs documenting morphology of H69 cells exposed to *H. pylori* at 0, 10, and 100 bacilli per cholangiocyte, respectively (left to right). The cellular appearance changed from an epithelial phenotype to a mesenchymal phenotype as evidenced by the loss of cell-cell contact, an elongated and spindle-shaped morphology, along with growth as individual cells by 24 h following exposure to *H. pylori*, in a dose-dependent manner. Scale bars, 5 μm (right), 20× magnification. The length-to-width ratio of single, isolated cells was determined to document elongation and scattering of the cell population (**D**,**E**). The number of elongated cells increased in a dose-dependent fashion in response to *H. pylori* (**F**). By contrast, the number of isolated, individual cholangiocytes, indicative of cell scattering, was also significantly increased in dose-dependent fashion (**G**). Data are presented as the mean ± standard error of three biological replicates. Means were compared using a one-way ANOVA. Asterisks indicate levels of statistical significance of experimental compared to control groups at 24 h; *, *p* ≤ 0.05; **, *p* ≤ 0.01; ***, *p* ≤ 0.001; ****, *p* ≤ 0.0001.

**Figure 2 pathogens-09-00971-f002:**
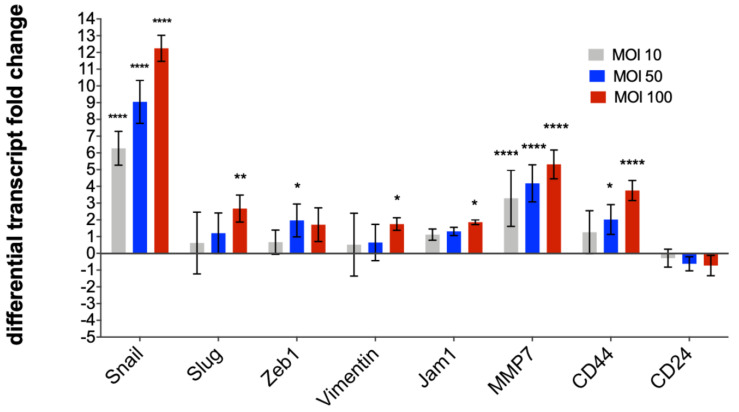
Differential transcript fold change of EMT-related and cancer stem cell marker genes after exposure to *Helicobacter pylori*. Messenger RNA expression of six EMT-related genes and two cancer stem cell markers were determined after 24 h of infection. Expression of Snail, Slug, vimentin, JAM1, MMP7, and CD44 increased in a dose-dependent fashion, whereas CD24 transcription did not change significantly. Expression of the regulatory transcriptional factor, Snail, was notably up-regulated by 6.27-fold ± 1.02-fold, 9.05-fold ± 1.28-fold, and 12.25-fold ± 0.78-fold at an MOI 10, 50, and 100, respectively. MMP7 expression was markedly up-regulated by 3.3- to 5.3-fold at each MOI. Expression of each of Slug, ZEB1, vimentin, and JAM1 was also up-regulated in a dose-dependent fashion. Transcription of the cancer stem cell marker CD44 was significantly up-regulated by 2.02-fold ± 0.88-fold at an MOI 50 and by 3.75-fold ± 0.60-fold at MO of 100, whereas significant change was not evident with CD24. Three biological replicates were carried out. The qPCR findings were normalized to the expression levels of GAPDH in each sample, with the mean ± S.D. values shown for the seven genes at an MOI of 10, 50, and 100, and compared using a two-way ANOVA multiple comparison with a 95% confidence interval of difference. *, *p* ≤ 0.05; **, *p* ≤ 0.01; ***, *p* ≤ 0.001; ****, *p* ≤ 0.0001.

**Figure 3 pathogens-09-00971-f003:**
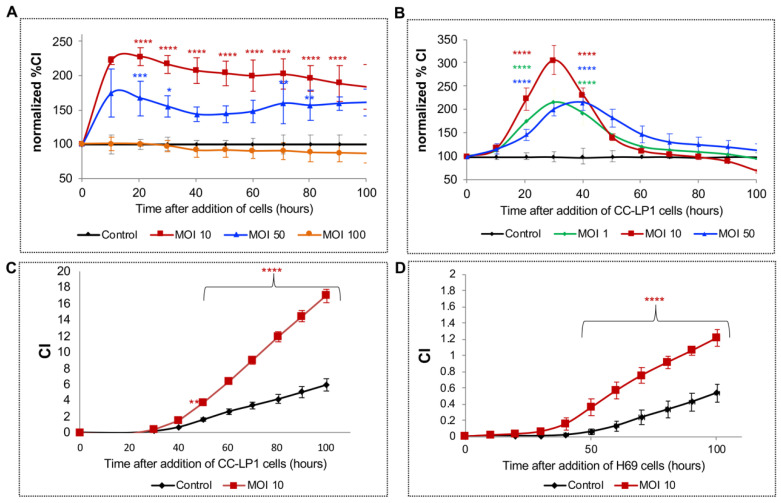
Exposing cholangiocytes to *H. pylori* induced migration and invasion through an extracellular matrix. H69 line cholangiocytes and CC-LP-1 line cholangiocarcinoma cells infected with *H. pylori* migrated through Matrigel, as monitored in real time over 100 h using CIM plates fitted to a xCELLigence DP system. Cell migration and invasion were monitored continuously for 100 h with the chemo-attractant in the lower chamber of the CIM plate. H69 cells infected with an MOI of 10 exhibited the fastest migration rate followed by cells exposed to MOI 50 *H. pylori*, whereas at an MOI of 100, cell migration was attenuated to a level comparable with non-infected control (**A**). In a similar fashion, the cholangiocarcinoma cell line CC-LP-1 showed the fastest migration when stimulated with an MOI of 10 *H. pylori* followed by an MOI of 50 *H. pylori*. Moreover, CC-LP-1 cells migrated significantly faster even at an MOI of 1 *H. pylori* (**B**). Invasion of the Matrigel extracellular matrix were compared between an MOI of 10 and non-infected cells. Invasion rates for CC-LP-1 and H69 cells significantly increased following exposure to *H. pylori* (**C**,**D**). *, *p* ≤ 0.05; **, *p* ≤ 0.01; ***, *p* ≤ 0.001; ****, *p* ≤ 0.0001.

**Figure 4 pathogens-09-00971-f004:**
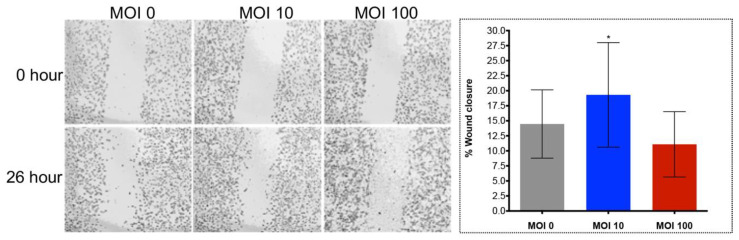
Wound healing in a two-dimensional in vitro assay revealed cell migration of H69 cholangiocytes exposed to *H. pylori*. Wound closure was significantly increased to 19.47% at an MOI of 10 of *H. pylori* (*, *p* ≤ 0.05) by a one-way ANOVA. An increase in wound closure was not apparent at an MOI of 100; 11.09% vs. control, 14.47%. Micrographs at 0 (top) and 26 (bottom hours; 5× magnification.

**Figure 5 pathogens-09-00971-f005:**
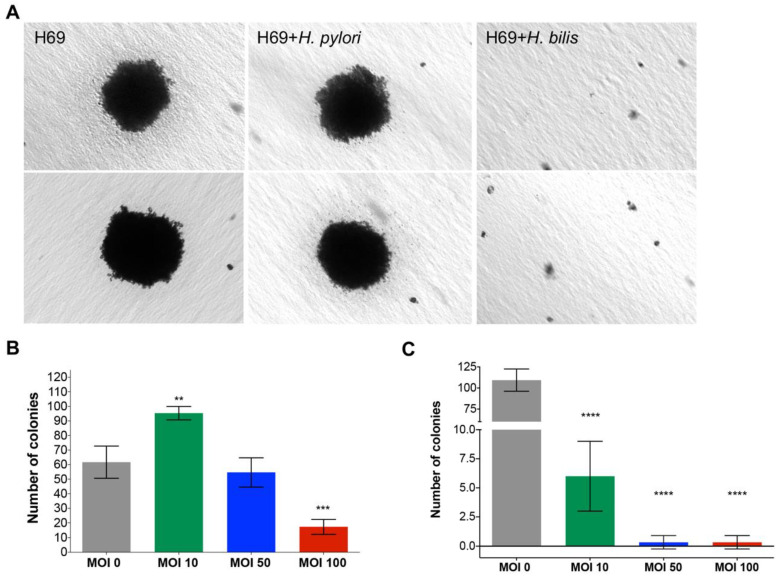
Anchorage-independent cell growth in soft agar revealed cellular transformation of cholangiocytes following exposure to *H. pylori*. Representative micrographs revealing the appearance of colonies of H69 cholangiocytes at 30 days following exposure to *H. pylori* and to *H. bilis*, as indicated (**A**). At an MOI of 10, the number of H69 cell colonies increased significantly (**, *p* ≤ 0.01), whereas they did not at an MOI of 50. By contrast, there was a significant decrease in colony numbers at an MOI of 100 when compared with the non-infected control cells (***, *p* ≤ 0.001) (**B**). Exposure of H69 cells to *H. bilis* resulted in markedly reduced numbers of colonies, in a dose-dependent fashion, indicating an inhibitory effect of *H. bilis* on anchorage-independent cell growth and/or cellular transformation of cholangiocytes (**C**). Three biological replicates were performed; mean ± S.E; (****, *p* ≤ 0.0001).

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
