# Peer review of "Infection with Helicobacter pylori Induces Epithelial to Mesenchymal Transition in Human Cholangiocytes"

_pathogens, 2020, doi:10.3390/pathogens9110971_

Round 1
Reviewer 1 Report
In this work the authors investigated the interactions between CagA+ H. pylori, the related species H. bilis and human cholangiocytes, on the basis of the evidence that infection with CagA+ H. pylori induced epithelial to mesenchymal transitition in the H69 cell line of human cholangiocytes.
The first sentence in the section discussion should be deleted because it is not appropriate in this context.
Author Response
1. The first sentence of the Discussion of the original version has been deleted.
2. Minor corrections have been made throughout the R1 version to improve, spelling, grammar, and syntax.
3. At large, the manuscript, and in particular the Discussion, has been revised extensively. Revised text is indicated using a red-colored font.
Reviewer 2 Report
In this paper, the autors investigated interactions between CagA+ H. pylori, the related species H. bilis, and human cholangiocytes. They found that infection with CagA+ H. pylori induced EMT in the H69 cell line of human cholangiocytes .
The paper is well written and overall merit.
I have some points:
1. Can the authors insert the characteristics of the H. pylori strains used in the materials and methods?
2. All experiments require a control with a Hp CagA- H. pylori in order to make comparison and the specific contribution of CagA.
Author Response
- Can the authors insert the characteristics of the H. pylori strains used in the materials and methods?
- All experiments require a control with a Hp CagA- H. pylori in order to make comparison and the specific contribution of CagA.
Authors' responses:
- More details have been provided in the Material and Methods section (red-colored font in the R1 version of the manuscript), lines 406-423:
Helicobacter pylori and Helicobacter bilis
Helicobacter pylori NCTC 11637 = CCUG 17874 = ATCC 43504 (Epsilon-proteobacteria), infraspecific name, strain NCTC 11637, from human gastric antrum [45], and Helicobacter bilis (ATCC® 49314™), originally from aborted sheep fetus, liver and fluids, from Brookings, South Dakota [38, 94, 95], were purchased from the American Type Culture Collection (ATCC) (Manassas, VA). Both were maintained on trypticase soy agar with 5% sheep blood for 72 to 96 hours [96] (Becton, Dickinson and Company, Franklin Lakes, NJ) under a microaerobic atmosphere established with the BD GasPak™ EZ container system and reagents (Becton Dickinson). The GasPak container with the plates inoculated with the bacteria was located within an orbital shake (MaxQ™ 4000 Benchtop Orbital Shaker, Thermo Fisher) and incubated at 37°C for 96 hours. At this point, each species of Helicobacter was harvested by scraping the colonies from the agar plate, and scrapings of the culture on the agar pale were resuspended in DMEM/F12 medium until an optical density at 600 nm of 1. 0 was reached, which corresponds to ~1ï‚´108 colony forming units (cfu)/ml [97-99]. Viability of the bacteria was confirmed by visual inspection for movement of the flagellated bacteria.
2. Control using CagA negative H. pylori. We concur. We have extensively revised the Discussion to address this aspect and to provide other relevant information.
Reviewer 3 Report
The authors of the manuscript by Thanaphongdecha et al. described the influence of H. pylori on human cholangiocytes.
The manuscript is well written although some information could be added to increase clarity of the presented study, including:
How many cells were measured in morphological alternation experiment,
How authors were able to measure length to width ration in H69 cell without addition of H. pylori, as the cell line does not grow as single separated cells?
According to EMT-associated-factors induced by exposure to H. pylori section CD24 is not changing expression after addition of H. pylori but authors mentioned at least twice in the text that the expression level decreased, why?
In the result section 2.4, why do we observe twice as many colonies in soft agar in the experiment with addition of H. bilis comparing to addition of H. pylori with MOI 0?
Discussion section is the weakest point of the manuscript. The authors did not discuss obtained results, but more summarize it and only pick the results that they describe. I did not find any information why MOI 100 of H. pylori usually did not show any influence on the tested cell lines.
Is it only Cag+ strain of H. pylori responsible for the observed changes and response from induced cell lines? The control using Cag- strain should be included in the presented study.
Section of materials and methods also should be improved. Could authors comment why Helicobacter that grew on solid medium needed gentle agitation?
Author Response
We thank the reviewer for her/his expert advice and recommendations. We have inserted the authors’ responses and revisions in blue ink below.
The authors of the manuscript by Thanaphongdecha et al. described the influence of H. pylori on human cholangiocytes.
The manuscript is well written although some information could be added to increase clarity of the presented study, including:
How many cells were measured in morphological alternation experiment,
Approximately 30, 60, and 70 cells - in total during the three biological replicates - at the MOI 0, 10 and 100 levels, respectively.
How authors were able to measure length to width ration in H69 cell without addition of H. pylori, as the cell line does not grow as single separated cells?
Under the growth conditions of the assay, as detailed in the Materials & Methods section, isolated H69 cells were observed occasionally at MOI = 0 (the control group) although the numbers increased in H. pylori at MOI 10 and even more at H. pylori MOI 100. Please see Figure 1, panels A, B, C and F. In like fashion, the length to width ratio, and numbers of cells with an elongate phenotype, increased in a dose dependent way (Figure 1G).
According to EMT-associated-factors induced by exposure to H. pylori section CD24 is not changing expression after addition of H. pylori but authors mentioned at least twice in the text that the expression level decreased, why?
The reviewer is correct. It was not correct for us to state that, and we have revised the manuscript to correct our error.
Whereas CD24 did marginally decrease the change was not statistically significant (Figure 2). We have revised the R1 version to clearly state that the expression of CD24 did not change significantly. Please see, for example, the Results section, line 165: ‘Transcription of Slug, ZEB1, vimentin and JAM1 also was up-regulated during bacterial colonization. CD44 was up-regulated whereas CD24 expression was not changed significantly, revealing a CD44+ high/CD24+ low phenotype in cells exposed to H. pylori…’
Also, please see the legend of Figure 2,
‘Figure 2. Differential transcript fold change of EMT-related and cancer stem cell marker genes after exposure to Helicobacter pylori. Messenger RNA expression of six EMT-related genes and two cancer stem cell markers were determined after 24 hours of infection. Expression of Snail, Slug, vimentin, JAM1, MMP7, and CD44 increased in a dose fashion whereas CD24 transcription did not change significantly. Expression of the regulatory transcriptional factor, Snail was notably up-regulated by 6.27±1.02-, 9.05±1.28- and 12.25±0.78-fold at MOI 10, 50, and 100, respectively. MMP7 expression was markedly up-regulated by 3.3 to 5.3 fold at each MOI. Expression of each of Slug, ZEB1, vimentin, and JAM1 also was up-regulated in a dose dependent fashion. Transcription of the cancer stem cell marker CD44 was significantly up-regulated by 2.02±0.88 fold at MOI 50 and by 3.75±0.60 fold at MO of 100 whereas significant change was not evident with CD24. Three biological replicates were carried out. The qPCR findings were normalized to the expression levels of GAPDH in each sample, with the mean ± S.D. values shown for the seven genes at each of MOI of 10, 50 and 100, and compared using a two-way ANOVA multiple comparison with a 95% confidence interval of difference’.
In the result section 2.4, why do we observe twice as many colonies in soft agar in the experiment with addition of H. bilis comparing to addition of H. pylori with MOI 0?
The representative micrographs shown in Figure 2.5 reflect the significant trends observed over several biological replicates. Assays with H. pylori and assays with H. bilis were not performed on the same day, using H9 cells cultured as needed to the desired level of confluence. The observation made by the reviewer reflects this practicality of setting us the assays on different days. And, as stated in the Materials & Methods section, line 541,‘Fresh medium was added every three or four days for 28 days or until the colonies had established, at which time the number of colonies of ³ 50 µm diameter was documented using a camera fitted to an inverted microscope (above), and quantified [119]’.
Discussion section is the weakest point of the manuscript. The authors did not discuss obtained results, but more summarize it and only pick the results that they describe. I did not find any information why MOI 100 of H. pylori usually did not show any influence on the tested cell lines.
We have expanded the Discussion to more fully address our findings. (However, the Discussion is already comparatively lengthy.) The third and fourth paragraphs of the Discussion, in particular, have been revised and expanded, and now address the issue of MOI 100:
‘AGS cells infected with H. pylori demonstrating the hummingbird phenotype also display early transcriptional changes that reflect EMT [28]. Here, H69 cells exposed to H. pylori exhibited upregulation of expression of Snail, Slug, vimentin, JAM1, and MMP7 in a dose-dependent fashion, changes that strongly supported the EMT of this informative cholangiocyte cell line. Likewise, Snail, an E-cadherin repressor, was markedly up-regulated, which indicated that this factor may be a key driver of EMT in cholangiocytes. CD44 expression was up-regulated in dose-dependent fashion, whereas CD24 was not significantly changed, revealing a CD44+/CD24-/low phenotype following infection. Colonization by H. pylori may not only induce EMT but also may contribute to stemness and malignant transformation of the cholangiocyte [37, 57].
‘Realtime cell monitoring revealed that the H69 and CC-LP-1 cells proliferated, and migrated in response to H. pylori. Moreover, 10 bacilli of H. pylori per biliary cell stimulated cellular migration and invasion by both H69 and CC-LP-1 through an extracellular basement matrix, a behavioral phenotype also characteristic of EMT. The responses by CC-LP-1 were even more marked than those of H69, which is not surprising given that CC-LP-1 is a cancer cell line. In addition, following exposure to H. pylori, H69 cells accomplished anchorage-independent cell growth in soft agar, a phenotype indicative of the escape from anoikis and characteristic of metastasis [58]. The numbers of colonies of H69 cells in soft agar increased significantly following exposure to H. pylori at MOI of 10. By contrast, observations of contact independent cell growth and also growth responses by H69 cells to H. bilis as monitored and quantified using the RTCA approach, revealed that both the numbers of colonies of H69 in soft agar and cell growth during the RTCA assay decreased. The responses indicated that H. pylori strain NCTC 11637, but not H. bilis, displays the potential to induce neoplastic changes in cholangiocytes in like fashion to the gastric epithelia. Overall, the pro-carcinogenic changes induced by H. pylori but not by H. bilis was not unexpected, given that H. pylori is a biological carcinogen, even though H. bilis colonized the intestines and hepatobiliary tract, has been associated with hepatobiliary disease including multifocal chronic hepatitis [39] and biliary tract malignancies [59]. Last, despite to clear stimulatory impact on the cholangiocytes of exposure of H. pylori, for example at MOI 10, the physiological response was generally dose dependent, cellular migration, wound healing, and contact independent colony growth were impeded or even blocked compared to enhanced growth stimulated by MOI 10. Dose dependent responses as well as negative impacts at higher MOI and/or higher concentrations of virulence factors of H. pylori have been reported during infection with H. pylori [60-64].’
Is it only Cag+ strain of H. pylori responsible for the observed changes and response from induced cell lines? The control using Cag- strain should be included in the presented study.
We have revised the Discussion extensively, including revision on this issue. Three new paragraphs of the R1 version of the Discussion directly address this point, including these two 'limitations' paragraphs:
Our report has some limitations. This study of the colonization of the human cholangiocyte by H. pylori was undertaken with the NCTC 11637 strain of Helicobacter pylori, a CagA positive infraspecific strain of the pathogen. We observed that exposure of H69 cells to H. pylori-induced EMT, cell proliferation and invasion, wound closure, and other responses. The report provides novel, informative findings but it is preliminary in scope with respect to the investigation of the impact of H. pylori on the biliary tract. Deeper investigation using other genotypes of H. pylori along with cognate deletion mutants would be appropriate in future studies in order to define specific virulence factors of H. pylori that underpin the phenotypic responses described here. The H. pylori strain NCTC 11637 is CagA positive, and given the established and well-characterized role of CagA in malignant transformation of the gastric epithelium [29, 82-84], assays that include additional CagA-positive strains such as P12 [85], and the inclusion of a CagA negative and/or cagPAI-dysfunctional strain such as SS1 [86-88], would explore the specific contribution of the CagA virulence factor to the EMT and other responses observed in the H69 line cholangiocyte. Specifically, induction of the hummingbird phenotype in gastric epithelial cells by CagA is dependent on the number and type of its EPIYA phosphorylation site repeats [83, 89]. Moreover, to confirm that the CagA virulence factor of H. pylori strain NCTC 11637 induced the elongate, hummingbird-like appearance of the H69 cholangiocyte, a demonstration that H69 cells could be infected also by another CagA positive H. pylori, and that the cagPAI cag pathogenicity island encoded type IV secretory system was functional and active during the infection, as established by detection of phosphorylated CagA [90-92], would be required.
Although H. pylori strain NCTC 11637 is CagA positive, other virulence factors of H. pylori may underlie the phenotypic changes in the H69 cells given that some other virulence factors of H. pylori induce phenotypic changes in gastric epithelia similar to those observed here in cholangiocytes. Notably, the outer membrane antigen, lipoprotein 20 (Lpp20) [93] promotes cell migration and down-regulation of E-cadherin in gastric cancer cells, responses involved with EMT [94]. Lpp20 also stimulates cell proliferation. As with investigation of the role of CagA discussed above, similar investigation might be undertaken to confirm a role for Lpp20, and indeed for other virulence factors.
Section of materials and methods also should be improved. Could authors comment why Helicobacter that grew on solid medium needed gentle agitation?
In like fashion to our responses to reviewer no. 2, the Materials & Methods section has been revised and upgraded. For example, please line 410 and the following sentences:
‘Helicobacter pylori and Helicobacter bilis
Helicobacter pylori NCTC 11637 = CCUG 17874 = ATCC 43504 (Epsilon-proteobacteria), infraspecific name, strain NCTC 11637, from human gastric antrum [45], and Helicobacter bilis (ATCC® 49314™), originally from aborted sheep fetus, liver and fluids, from Brookings, South Dakota [38, 94, 95], were purchased from the American Type Culture Collection (ATCC) (Manassas, VA). Both were maintained on trypticase soy agar with 5% sheep blood for 72 to 96 hours [96] (Becton, Dickinson and Company, Franklin Lakes, NJ) under a microaerobic atmosphere established with the BD GasPak™ EZ container system and reagents (Becton Dickinson). The GasPak container with the plates inoculated with the bacteria was located within an orbital shake (MaxQ™ 4000 Benchtop Orbital Shaker, Thermo Fisher) and incubated at 37°C for 96 hours. At this point, each species of Helicobacter was harvested by scraping the colonies from the agar plate, and scrapings of the culture on the agar pale were resuspended in DMEM/F12 medium until an optical density at 600 nm of 1. 0 was reached, which corresponds to ~1´108 colony forming units (cfu)/ml [97-99]. Viability of the bacteria was confirmed by visual inspection for movement of the flagellated bacteria. ‘
Could authors comment why Helicobacter that grew on solid medium needed gentle agitation?
Helicobacter grows on solid medium with or without gentle agitation; however, gentle agitation is not needed.
Round 2
Reviewer 2 Report
I review the revised version and
I believe the manuscript has been significantly
improved and now warrants publication in Pathogens.
Author Response
We thank reviewer no. 2 for the positive comments.
As s/he recommended ((x) English language and style are fine/minor spell check required), we re-checked the manuscript for typographical and grammatical errors.
1. We prepared several minor issues with punctuation.
3. And revised lines 63 to 67 of the Abstract,
'Analysis to quantify cellular proliferation, migration and invasion in real-time by both H69 cholangiocytes and CC-LP-1 line of cholangiocarcinoma cells using the xCELLigence approach and Matrigel matrix revealed that exposure to ³10 H. pylori bacilli per cell stimulated migration and invasion by the cholangiocytes. In addition, 10 bacilli of H. pylori stimulated contact-independent colony establishment in soft agar'.